# Hypoxia as a Stimulus for the Maturation of Meniscal Cells: Highway to Novel Tissue Engineering Strategies?

**DOI:** 10.3390/ijms22136905

**Published:** 2021-06-27

**Authors:** Valentina Rafaela Herrera Millar, Laura Mangiavini, Umberto Polito, Barbara Canciani, Van Thi Nguyen, Federica Cirillo, Luigi Anastasia, Giuseppe Maria Peretti, Silvia Clotilde Modina, Alessia Di Giancamillo

**Affiliations:** 1Department of Veterinary Medicine, University of Milan, 26900 Lodi, Italy; valentina.herrera@unimi.it (V.R.H.M.); umberto.polito@unimi.it (U.P.); silvia.modina@unimi.it (S.C.M.); 2IRCCS Istituto Ortopedico Galeazzi, 20161 Milan, Italy; laura.mangiavini@unimi.it (L.M.); canciani.barbara@hsr.it (B.C.); nguyenthivan.sha49@gmail.com (V.T.N.); giuseppe.peretti@unimi.it (G.M.P.); 3Department of Biomedical Sciences for Health, University of Milan, 20133 Milan, Italy; 4Stem Cells for Tissue Engineering Laboratory, IRCCS Policlinico San Donato, 20097 San Donato Milanese, Italy; federica.cirillo@grupposandonato.it (F.C.); anastasia.luigi@unisr.it (L.A.); 5Università Vita-Salute San Raffaele, 20132 Milan, Italy

**Keywords:** glycosaminoglycans, HIF-1α, hypoxia, meniscus, pig, fibro-chondrocytes

## Abstract

The meniscus possesses low self-healing properties. A perfect regenerative technique for this tissue has not yet been developed. This work aims to evaluate the role of hypoxia in meniscal development in vitro. Menisci from neonatal pigs (day 0) were harvested and cultured under two different atmospheric conditions: hypoxia (1% O_2_) and normoxia (21% O_2_) for up to 14 days. Samples were analysed at 0, 7 and 14 days by histochemical (Safranin-O staining), immunofluorescence and RT-PCR (in both methods for SOX-9, HIF-1α, collagen I and II), and biochemical (DNA, GAGs, DNA/GAGs ratio) techniques to record any possible differences in the maturation of meniscal cells. Safranin-O staining showed increments in matrix deposition and round-shape “fibro-chondrocytic” cells in hypoxia-cultured menisci compared with controls under normal atmospheric conditions. The same maturation shifting was observed by immunofluorescence and RT-PCR analysis: SOX-9 and collagen II increased from day zero up to 14 days under a hypoxic environment. An increment of DNA/GAGs ratio typical of mature meniscal tissue (characterized by fewer cells and more GAGs) was observed by biochemical analysis. This study shows that hypoxia can be considered as a booster to achieve meniscal cell maturation, and opens new opportunities in the field of meniscus tissue engineering.

## 1. Introduction

Regenerative medicine is trying to achieve the goal of replacement or regeneration of the meniscus, as it plays a fundamental role in the biomechanics of the knee joint [1]. Despite great efforts during the last few years, the different techniques adopted to replace or regenerate meniscus still present some criticisms and pitfalls [2,3,4]. The main drawbacks are cell tendency to dedifferentiate during the in vitro culture (with a decrease in collagen II and aggrecan), the poorer biomechanical properties of the meniscal substitutes and the lack of integration between the cultured cells and the scaffolds [2,3]. These side effects may be caused by an improper technique or by inadequate culture conditions of meniscal tissue or cells. Therefore, in the last few years, several experiments have been focused on the modulation of the physiological environment of the meniscus. For example, the articular space is highly hypoxic (1–9% O_2_) [5,6,7], even if cell cultures are usually performed in normoxic conditions (≈21% O_2_). Thus, menisci (in particular, the inner/avascular zone) and cartilage are generally cultured in a hyperoxic environment, which may somewhat false the results and may explain some discrepancies that have been observed in the passage from in vitro to in vivo. Cartilaginous cells can survive in the inhospitable hypoxic environment thanks to the hypoxia-inducible factor-1 alpha (HIF-1α), an essential factor for chondrocyte survival and cartilage homeostasis [8,9,10]. Hypoxia-inducible factors (HIFs) are oxygen-sensitive transcriptional complexes constituted by α- and β-subunits that activate several pathways regulating cellular proliferation and metabolism [8,9,10,11,12]. In normoxia, the HIF-1α transcriptional subunit is recognized and hydroxylated on specific proline-residues by HIF prolyl-4-hydroxylases; the hydroxylated subunit is then recognized by the E3 ubiquitin ligase von Hippel–Lindau (VHL) that targets HIF-1α to the proteasome for degradation [8,9]. Under lower oxygen concentration conditions (i.e., hypoxia), HIF-1α stabilizes, translocates to the nucleus, and binds to HIF-1β to exert its transcriptional activity [9,10,13]. Most studies on chondrogenesis, either in pellet or scaffold cultures, indicate an increase in chondrogenic gene expression (*SOX-9*, *COL2A1* and *ACAN*) [14] and matrix formation under hypoxia [14]. In particular, HIF-1α interacts with SOX-9 to up-regulate collagen type II (*COL2A1*) and aggrecan (*ACAN*) genes (both characteristic of the cartilaginous phenotype), whereas collagen type X (*COL10A1*), a marker of endochondral ossification, is down-regulated in these conditions [14]. Hypoxic culture (from 1% to 13% O_2_) is also associated with similar results in meniscus cultures, with an enhancement of fibro-chondrocytic phenotypical traits in human meniscal cell aggregates [15]. In human meniscus cell aggregates, hypoxia induces an increment of *SOX-9*, linked to the post-transcriptional effect of HIF-1α and, consequently, an increase in collagen type II deposition, both characteristics of cartilaginous tissue and signs of maturation of fibro-chondrocytes in meniscus [15].

The application of hypoxia may be a fundamental step in the tissue culture of meniscus, avoiding the dedifferentiation and promoting the subsistence of phenotypic features in meniscal samples, as suggested in previous works on cartilage and meniscal cells cultures [14,15,16,17]. Nevertheless, the effect of this environmental condition upon the whole meniscal tissue, its cellular phenotypes and extracellular matrix composition, is still a matter of interest, and the application of hypoxia on an immature tissue, composed by committed but not already functional cells, has not been yet evaluated. In this study, we have assessed the pure effect of hypoxia (1% O_2_) upon differentiation, proliferation and endogenous activation of committed cells within the whole neonatal meniscal tissue, characterized by the native relationship between cells and extracellular matrix.

## 2. Results

### 2.1. Morphological Evaluation—Histochemistry

Menisci were analysed by histochemical staining to evaluate the general morphology and matrix deposition of cells at each time point and under the two different oxygen tensions (Figure 1). Initially, the neonatal meniscal cells presented a fibroblast-like phenotype without matrix deposition (Figure 1A inner zone). Both the inner (Figure 1A) and the outer (Figure 1B) zones of the meniscus appeared vascularized and featured fibrous tissue (Figure 1B, thin arrow). Moreover, blood vessels were present both in the inner and outer zone at each time point, and in both conditions (representative region of outer zone in Figure 1B,D,F, thick arrows). After seven days in culture (Figure 1C,E inner zone; D,F outer zone), the cells acquired a round shape, mirroring a fibro-chondrocyte-like phenotype, both in normoxic (Figure 1C inner zone, asterisk) and hypoxic (Figure 1E inner zone, arrowheads) conditions. Fibroblast-like cells were observed in the outer zone both in normoxic and hypoxic treatment (Figure 1D,F, thin arrows). At the longest time point (T2), the extracellular matrix was predominantly found in the inner zone (Figure 1G,I asterisk), both in normoxic (Figure 1G) and hypoxic (Figure 1I) conditions, where round-shaped cells were present. On the other hand, the outer zone was still characterized by fibroblast-like cells (Figure 1H,J, double asterisks).

### 2.2. Morphological Evaluation—Immunofluorescence

Histochemical results were confirmed by double immunofluorescence (Figure 2). At the initial time point, the meniscal tissue was composed by cells with elongated fusiform nuclei (Figure 2A, blue) expressing the SOX-9 (Figure 2B, green) protein, but they did not express COL2A1 (Figure 2C, merge). At seven days (T1), under both oxygen tension conditions, the nuclei acquired a more rounded shape (Figure 2D,G) and they expressed both SOX-9 (Figure 2E,H, green) and COL2A1 (Figure 2E,H, red). Collagen type II was only present in the nuclei under normoxia (Figure 2E, nuclear co-expression of COL2A1 and SOX-9, yellow), whereas in hypoxic conditions, COL2A1 was expressed both within the nuclei (Figure 2H, nuclear co-expression of COL2A1 and SOX-9, yellow) and in the extracellular matrix (Figure 2H, red). At fourteen days (T2), we observed two different patterns linked to the oxygen tension. Menisci under normoxia were characterized by cells that lose their round shape in favour of a more elongated form (Figure 2J, blue). Furthermore, COL2A1 expression was abolished (Figure 2K, red). SOX-9 greatly decreased, and it was limited to the (few) still rounded nuclei (Figure 2K, green). On the contrary, hypoxic-cultured menisci presented cells that still preserved their round shape nuclei (Figure 2M) and a well-defined extracellular deposition of collagen II (Figure 2N, red), not strictly linked to the expression of SOX-9 (Figure 2N, green or yellow when co-expressed).

### 2.3. Biochemical Analysis 

The biochemical analysis confirmed the histological findings, as hypoxic cultured menisci displayed a higher GAGs concentration at T2 (Figure 3A), though without a significant difference. Moreover, GAGs production was significantly increased in both normoxic and hypoxic samples at T2, when compared with T0 (*p* < 0.05 both comparisons). DNA quantification (Figure 3B) reflected the number of cells, although hypoxic samples were characterized by a lower cellularity both at T1 and T2 compared to T0, with no statistically significant differences observed. The ratio between GAGs production and cellularity (GAGs/DNA; Figure 3C) reflected the grade of maturation of the tissue. Mature tissue is characterized by fewer cells producing a higher amount of matrix, thus an increased GAGs/DNA ratio reflects a more mature tissue. Indeed, T0 showed the lowest value, while hypoxic menisci displayed the higher GAGs/DNA ratio at T2 (T0 vs. T2 *p* < 0.05). 

### 2.4. Real-Time PCR Assay: COL2A1, COL1A1, SOX-9, HIF-1α, ACAN

To evaluate the effect of the two oxygen tensions, different genes linked to the maturation process of meniscal tissue and to the adaptive response to hypoxia were evaluated. COL2A1 (Figure 4A), a marker of meniscal tissue maturation, was up-regulated in the hypoxic culture (p < 0.01): a higher significant expression was observed in hypoxia at T2 compared with T1 and to normoxia meniscal samples (p < 0.01, all comparisons). COL1A1 (Figure 4B) was down-regulated in hypoxia cultured menisci at both experimental time points compared to normoxic samples, but no statistical differences among groups was observed. The SOX-9 (Figure 4C) gene was up-regulated in the hypoxic meniscus after seven days of culture, though without statistical significance if compared with normoxia. Similarly to SOX-9, HIF-1α (Figure 4D) was up-regulated in the hypoxic meniscus after seven days of culture with no statistical significance. However, no statistically significant differences in HIF-1α expression were found among groups at all timepoints. ACAN expression decreased at both T1 and T2 in the two experimental groups, albeit slightly higher in the hypoxic group (no statistically significant difference was observed) (Figure 4E).

## 3. Discussion

This study aims to analyse the role of hypoxia in the development and maturation of neonatal meniscal tissue. The full weight-bearing leads to some modifications of meniscal vascularization, innervation and cellular density [18,19]. These modifications are strictly linked to the achievement of the different cellular phenotypes that characterize the mature meniscus: fibroblast-like cells in the outer zone, and fibrochondrocyte-like cells concentrated in the inner zone [2]. Hypoxia may mimic the effect of biomechanics compression upon vessels described during the physiological development of the tissue after the start of full load-bearing gait [18]. Indeed, during meniscal development, vascularization, innervation and cellular density progressively decrease in the inner zone, which becomes almost completely avascular and aneural, whereas the outer third of the meniscus preserves its original vascularization [18,19]. The cellular adaptation to lower oxygen tension is mediated by the hypoxia-inducible factor-1α, which contributes to the differentiation of fibroblast-like cells in fibrochondrocyte-like cells. Fibrochondrocyte-like cells are predominant in the inner zone and can survive in the hypoxic environment [2,20]. In this study, we replicated this harsh environment using a hypoxia chamber, a specific apparatus that allows for controlling the oxygen tension present in the tissue culture environment. We analysed neonatal meniscus cultures in hypoxia or normoxia conditions. Histochemical results showed an increase in GAGs production after 7 and 14 days, especially in hypoxic samples. These cells also displayed a more mature phenotype, characterized by a higher GAGs production and by fewer cells with a fibrochondrocyte-like shape. These changes were also confirmed by a significantly higher GAGs/DNA ratio in the hypoxic cells. Moreover, the immunofluorescence staining for COL2A1, the final product of meniscal cells maturation, increased under the hypoxic stimulus starting from 7 days of culture. At this early time point, COL2A1 was co-expressed with SOX-9 in the nucleus. At the longest time point, the COL2A1 expression pattern changed, resulting in a homogenous extracellular deposition. Thus, hypoxia exerts a positive role on meniscus maturation, primarily on the collagen network and, secondarily, on the GAGs production; the biochemical evaluation did not show a significant difference in GAGs quantity between the two experimental conditions, and the *ACAN* gene expression revealed no significant differences. Similar histological and biochemical results were found in human cells aggregates cultured in vitro with fibroblast growth factor-2 under both environmental conditions [16]. GAGs are strictly related to the ability of the meniscus to sustain compressive loads. Thus, the lack of any significant differences between hypoxia- and normoxia-exposed samples may be linked to the absence of biomechanical stimuli in the in vitro culture. Gray et al. observed that oxygen delivery is essential in the first phase of meniscus development since the absence of gait loading and movement does not allow the correct distribution of nutrients from synovial fluid to meniscal cells [18]. Hypoxia represents a more powerful stimulus than dynamic compressive loading in promoting chondrogenesis in human cartilaginous cells [14]; consequently, this environmental condition may also favour meniscus development. Indeed, hypoxia may have similar effects to compression during the meniscus development, inducing the morphological adaptation of the mature meniscus, characterized by the expression of collagen type II and, at a later time, the production of GAGs. In our study, we used neonatal cells, which were committed to assume a fibrochondrocyte-like phenotype, but they were still immature; therefore, the temporal production and distribution of GAGs may have been delayed. Thus, the analysis of longer time points may reveal a possible hypoxia effect on GAGs production.

RT-PCR analysis evaluates the timely expression of the different meniscal genes, and it allows for a chronological track of the hypoxia role on meniscal maturation. Several authors demonstrated that the HIF-1α stabilization under low oxygen tension up-regulates *SOX-9* and *COL2A1* expression at 7 days and 14 days, respectively [14,15,16,21]. This increase, associated with a more rounded cellular shape, represents a recognized feature of meniscal maturation. Notably, *HIF-1α* gene expression decreases in our samples cultured in hypoxia for 14 days. Physiologically, mature meniscal cells reside in a hypoxic environment and are surrounded by an extracellular matrix [15,17,22]. Mature meniscal cells may present a higher resistance to hypoxia unlike the immature cells; thus, they are less dependent on HIF-1α adaptive effect, as shown in our samples at the longest time point. Moreover, Adesida et al. demonstrated that mature meniscal cells harvested from the less hypoxic outer region responded more to hypoxia compared with cells collected from the inner avascular zone [15]. In the present study, we did not separate the two regions due to the described weak regionalization of the meniscus in the initial stage of development [2]. Furthermore, species-specific differences in the composition of the native menisci may also have a role in these findings. Unlike humans, pigs are able to walk from the first day after birth [18]. 

Moreover, HIF-1α is not the only pathway involved in chondrogenic differentiation [23]. Several growth factors such as basic fibroblast growth factor, transforming growth factor -β3, and insulin growth factor-1 regulate this process independently from HIF-1α [16,17,24]. Thus, the effects of these alternative pathways on meniscal maturation should be considered and investigated in further studies. 

In the present study, hypoxia preserved the phenotypical characteristics of a mature meniscus avoiding the typical dedifferentiation observed in normoxic conditions. Indeed, dedifferentiation features, such as a decreased collagen type II expression in favour of collagen type I, are present in our normoxic samples, more evidently at the longest time point. Previous studies have demonstrated hypoxia effects both in aggregate meniscal cells and in cartilaginous samples [14,15,16,17]. However, to our knowledge, this is the first study that evaluates the effect of hypoxia on the whole meniscal tissue at a very early stage of differentiation. Thus, we analysed the pure effect of hypoxia upon a committed cells population within its native extracellular matrix, without the application of other stimuli. In conclusion, our results may represent a starting point towards novel tissue engineering strategies. 

## 4. Materials and Methods

### 4.1. Study Design

Neonatal menisci (n: 90) were collected from stillbirth pigs provided by a local breeding farm. The Ethics Committee of the University of Milan (OPBA, 58/2016) approved the use of cadavers for research purposes; furthermore, all the animals were deceased for causes not related with the present study. Eighteen menisci were analysed as a common starting point and the remaining 72 were split into two groups each of 36 menisci: hypoxia, i.e., with a tension of oxygen of 1%, and normoxia, i.e., normal (atmospheric) oxygen conditions (≈21% O_2_), as described in Figure 5. Menisci were stored within 6-well plates in a static culture with Dulbecco’s modified Eagles medium (DMEM) supplemented with 20% fetal bovine serum (FBS), 1000 unit/mL of penicillin/streptomycin and 25 mg/mL fungizone [25,26]. Hypoxia group’s menisci were maintained in a controlled atmosphere chamber with low tension of oxygen (1% O_2_), at 37 °C, changing the medium every 3 days, collecting samples at the midway and final time point (7 and 14 days, respectively). Normoxic samples were cultured under standard conditions in an atmosphere condition at approximately 21% of O_2_ at 37 °C. Eighteen menisci were analysed at each time point (T0, T7, and T14). Samples were examined by morphological analysis (histochemistry and immunofluorescence), biochemical analysis and real time PCR techniques (n: 6, per each group and technique). No differences regarding medial and lateral menisci were considered.

### 4.2. Morphological Evaluation—Histochemistry 

Samples were fixed in 10% buffered formalin (Bio-Optica, Milan, Italy) for 24 h, dehydrated and embedded in paraffin (total number of specimens: 30; 6 per each treatment and time point). Longitudinal whole meniscus sections (4 µm thickness) were analysed by Safranin-O staining (SO) to detect the presence of GAGs in the matrix, and to describe the meniscal structure [27].

### 4.3. Morphological Evaluation—Immunofluorescence

Immunofluorescence was applied to examine the expression and possible co-localization of specific proteins (collagen type II and SOX-9). After rehydration, heat-induced antigen retrieval was performed as previously described [28,29]. After washing three times in PBS (pH 7.4), sections were incubated with the first-step primary antiserum, 1:50 SOX-9 (Abcam, Cambridge, UK) for 24 h at 18–20 °C, then washed in PBS, and subsequently treated with the Avidin–Biotin blocking kit solution (Vector Laboratories Inc., Burlingame, CA, USA). The sections were then washed in PBS for 10 minutes and incubated with a solution of goat biotinylated anti-rabbit IgG (Vector Laboratories Inc.), 10 µg/ml in Tris-buffered saline (TBS) for 1 h at 18–20 °C. After rinsing twice in PBS, the sections were treated with Fluorescein–Avidin D (Vector Laboratories Inc.), 10 µg/ml in NaHCO_3_, 0.1 M, pH 8.5, 0.15 M NaCl for 1 h at 18–20 °C. For the second step of the double immunofluorescence procedure, sections were treated in a 2% hyaluronidase solution at room temperature for 30 min. The slides were subsequently treated with 1:50 anti-collagen II antiserum (Chondrex Inc., Redmond, WA, USA). Sections were rinsed in TBS for 10 min. and incubated with 10 µg/ml goat biotinylated anti-mouse IgG (Vector Laboratories Inc.) for 1 h at 18–20 °C. The sections were then washed twice in PBS, and treated with Rhodamine–Avidin D (Vector Laboratories Inc.), 10 µg/ml in NaHCO_3_, 0.1 M, pH 8.5, with 0.15 M NaCl for 1 h at 18–20 °C. Finally, slides with tissue sections were embedded in Vectashield Mounting Medium (Vector Laboratories Inc.) and observed using a Confocal Laser Scanning Microscope (FluoView FV300; Olympus). The immunofluororeactive structures were excited using Argon/Helio–Neon–Green lasers with excitation and barrier filters set for fluorescein and rhodamine. Images containing superimposition of fluorescence were obtained by sequentially acquiring the image slice of each laser excitation or channel. In a double immunofluorescence experiment, the absence of cross-reactivity with the secondary antibody was verified by omitting the primary antibody during the first incubation step.

### 4.4. Biochemical Analysis 

For each experimental group (0 days, hypoxic and normoxic samples at 7 and 14 days), 6 specimens were processed (*n* = 30). The samples for biochemical evaluation were digested in papain (Sigma-Aldrich, St. Louis, MO, USA) for 16–24 h at 60 °C; the digestion solution was composed of 125 µg/mL of papain (Sigma-Aldrich) in 100 mM sodium phosphate, 10 mM sodium EDTA (Sigma-Aldrich), 10 mM cysteine hydrochloride (Sigma-Aldrich), 5 mM EDTA adjusted to pH 6.5 and brought to 100 mL of solution with distilled water. The digested samples were stored at −80 °C. Aliquots of the papain digests were assayed separately for proteoglycan and DNA contents. Proteoglycan content was estimated by quantifying the amount of sulphated glycosaminoglycans using the 1,9-dimethylmethylene (DMMB) blue dye binding assay (Polysciences Inc., Washington, PA, USA) and a microplate reader (wavelength: 540 nm). The standard curve for the analysis was generated using bovine trachea chondroitin sulphate B (Sigma-Aldrich). DNA content was evaluated with the Quant-iT Picogreen dsDNA Assay Kit (Molecular Probes, Inc., Eugene, OR, USA) and a fluorescence microplate reader and standard fluorescein wavelengths (excitation 485 nm, emission 538 nm, cut-off 530 nm). The standard curve for the analysis was generated using bacteriophage lambda DNA supplied with the kit.

### 4.5. Real-Time PCR Assay 

The tissues were homogenized and extracted by using RNeasy Mini Kit (Qiagen, Hilden, Germany). Quantity and quality of RNA were determined by using Nanodrop 8000 (ThermoFisher Scientific, Portland, OR, USA). RNA was reverse transcribed by using ImProm II reverse Transcription System (Promega, Milan, Italy). Amplification of cDNA was performed by using PowerUp SYBR master mix (ThermoFisher Scientific) on 7500 Fast Realtime PCR System (Applied Biosystems, Foster City, CA, USA). The sequences of primers were listed on Table 1. The reactions were performed in three stages: holding stage initializing at 50 °C for 20 s, then 95 °C for 10 min; cycling stage at 95 °C for 15 s, then 60 °C for 1 m. The cycling stage was repeated for 40 cycles. Finally, in the melt curve stage, the reactions were set at 95 °C for 15 s, followed by 60 °C for 1 m, 95 °C for 30 s and 60 °C for 15 s. Data were analysed according to a comparative method where data were presented as fold change (2^−ΔΔCt^ value) with ∆Ct = [Ct (gene of interest)-Ct (beta-actin)] and ∆∆Ct = [(∆Ct at day n)—(∆Ct at day 0)], *n* = number of days of differentiation [30].

### 4.6. Statistical Analysis

Statistical analysis was performed with SAS statistical software (ver. 9.3, Cary, NC, USA). Data from the biochemical and RT-PCR analyses were analysed using 2-way ANOVA with time (0, 7 and 14 days) and treatment (normoxia or hypoxia) as main factors. The individual meniscus of each piglet was considered as the experimental unit. The data are presented as means with standard errors. Differences between means were considered significant at *p* < 0.05 and *p* < 0.01.

## 5. Conclusions

Our results suggest a positive role of hypoxia in the differentiation process of meniscal tissue. In particular, hypoxia may act as a booster for the production of a mature matrix and for the maintenance of the mature cell phenotype that characterizes the functional meniscus. These data open considerable opportunities in the field of meniscus tissue engineering. However, to achieve complete maturation that considers both collagen network and extracellular matrix production, the application of hypoxia alone may not be sufficient, considering a maximal culture time of 14 days). The combination with other stimuli, such as growth factors or biomechanical inputs, may be still necessary.

## Figures and Tables

**Figure 1 ijms-22-06905-f001:**
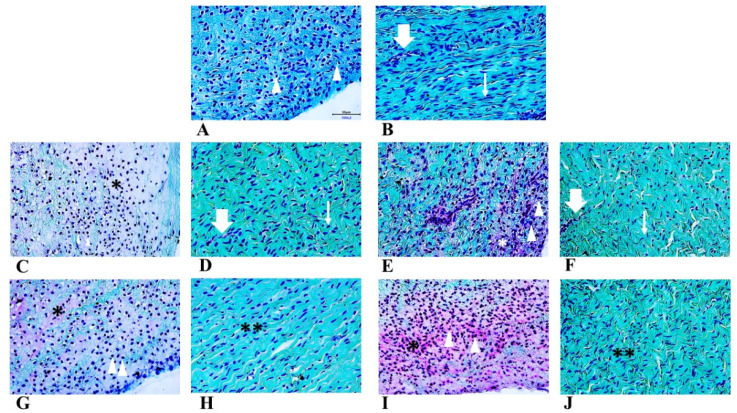
Safranin-O staining of neonatal meniscus (**A**: inner zone, **B**: outer zone). Tissue in normoxia for T1 (**C**: inner zone, **D**: outer zone) and T2 (**H**: inner zone, **I**: outer zone) timepoints. Tissue in hypoxia for T1 (**E**: inner zone, **F**: outer zone) and T2 (**G**: inner zone, **J**: outer zone) timepoints. Scale bar for all images: 50 µm. Arrows: blood vessels; arrowheads: fibro-chondrocytes like cells; thin arrows: fibroblast-like cells; asterisks: extracellular matrix deposition.

**Figure 2 ijms-22-06905-f002:**
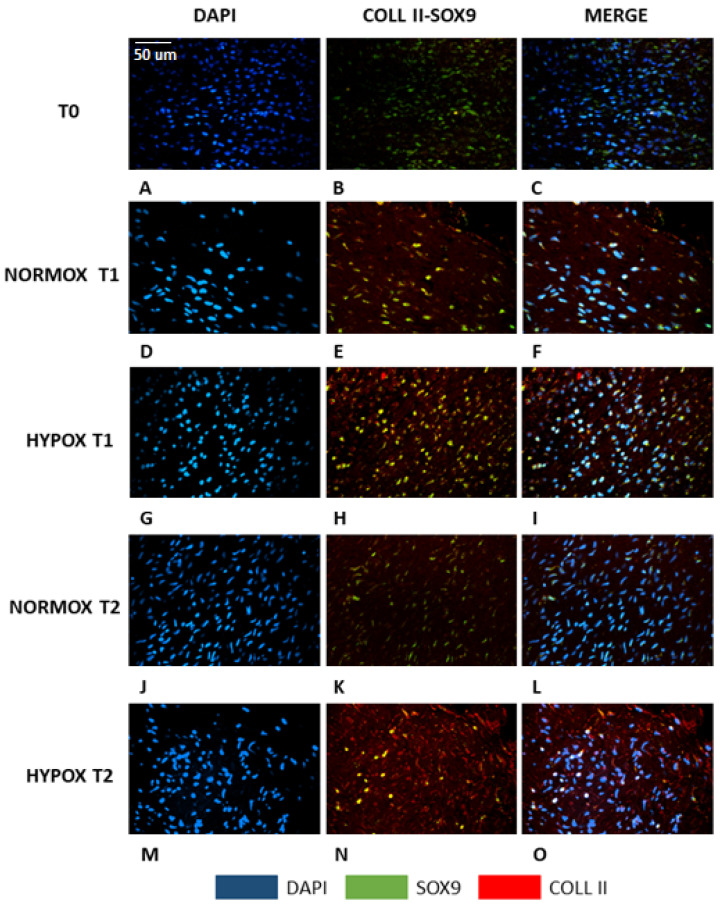
Double immunofluorescence of meniscal samples. **A**–**C**: native meniscus; **D**–**F**: meniscus cultured under normoxia for 7 days (T1); **G**–**I**: meniscus cultured under hypoxia for 7 days (T1); **J**–**L**: meniscus cultured under normoxia for 14 days (T2); **M**–**O**: meniscus cultured under hypoxia for 14 days (T2). Blue: DAPI; green: SOX-9; red: collagen type II; yellow: co-expression of SOX-9 and collagen type II. Scale bar for all images: 50 µm.

**Figure 3 ijms-22-06905-f003:**
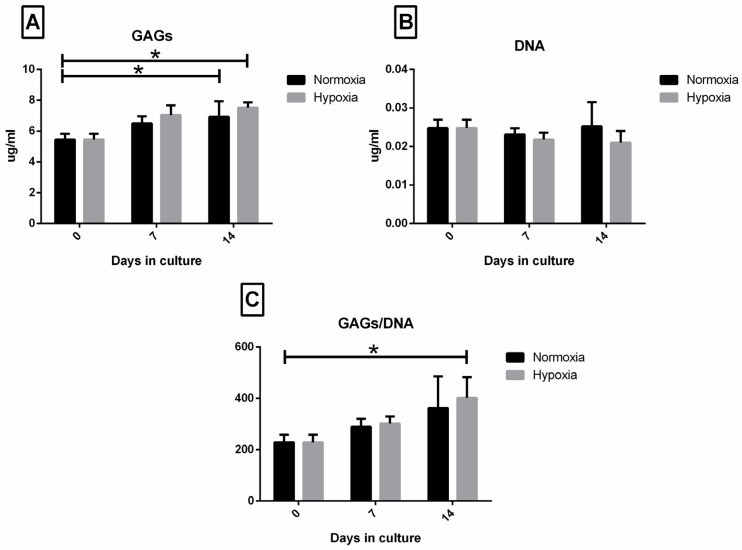
Biochemical analysis: GAGs (**A**), DNA (**B**) and GAGs/DNA ratio (**C**) analysis. Neonatal meniscus (T0) is compared to menisci cultured in normoxic and hypoxic conditions after 7 days (T1) and 14 days (T2). Values are expressed as mean ± SEM. Significant values are indicated with * when *p* < 0.05.

**Figure 4 ijms-22-06905-f004:**
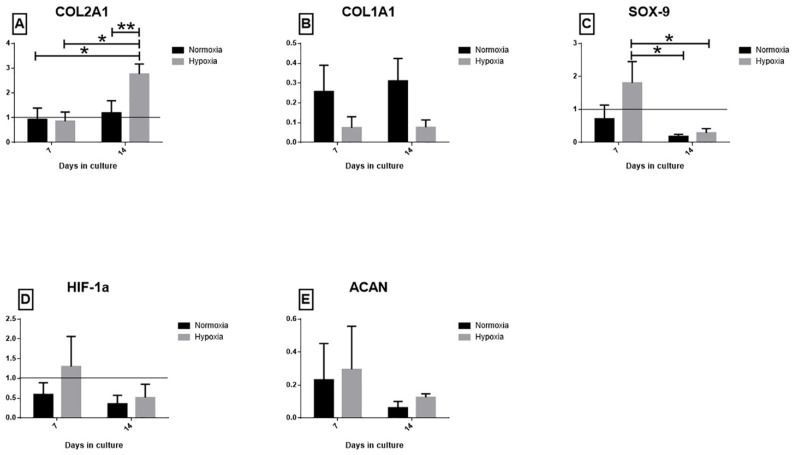
qPCR results. Gene expression of (**A**) COL2A1, (**B**) COL1A1, (**C**) SOX-9, (**D**) HIF-1α and (**E**) ACAN after 7 days (T1) and 14 days (T2). Data are presented as 2^−ΔΔCt^ ± SEM. Significant values are indicated with * when *p* < 0.05 and ** when *p* < 0.01.3.

**Figure 5 ijms-22-06905-f005:**
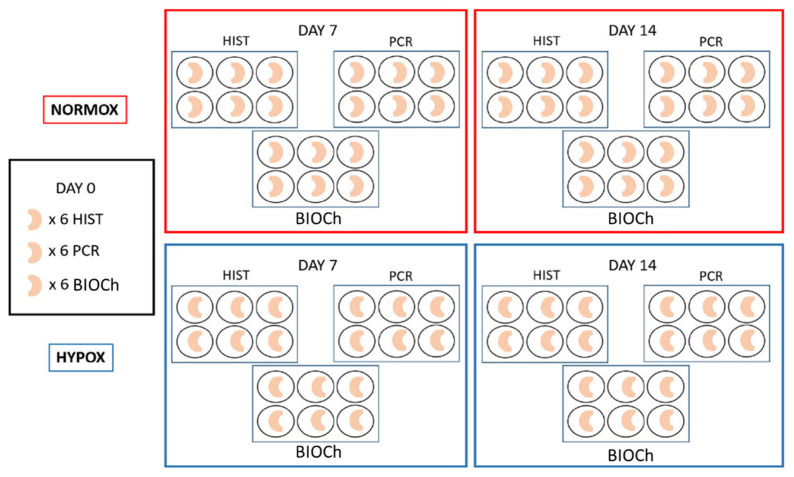
Schematic drawing of the study design.

**Table 1 ijms-22-06905-t001:** Primer sequences, F: Forward, R: Reverse.

No.	Gene	Sequence (5′–3′)	Amplicon Size (bp)	Reference
1	Β-ACT F	CAAGGAGAAGCTCTGCTACG	245	Kreinest et al., 2015
	Β-ACT R	AGAGGTCCTTCCTGATGTCC		
2	COL1A1 F	CCAACAAGGCCAAGAAGAAG	64	Kreinest et al., 2015
	COL1A1 R	ATGGTACCTGAGGCCGTTCT		
3	COL2A1 F	CACGGATGGTCCCAAAGG	102	Kreinest et al., 2015
	COL2A1 R	ATACCAGCAGCTCCCCTCT		
4	SOX-9 F	CCGGTGCGCGTCAAC	119	Kreinest et al., 2015
	SOX-9 R	TGCAGGTGCGGGTACTGAT		
5	HIF-1α F	AGGAATTATTTAGCATGTAGACTGCTGG	73	Gelse et al., 2008
	HIF-1α R	CATAACTGGTCAGCTGTGGTAATCC		
6	ACAN F	AAGGTTGCTACGGGG	113	
	ACAN R	GACCTCACCCTCCAT		

## Data Availability

Publicly available datasets were analysed in this study. This data can be found here: https://osf.io/76qup/?view_only=62c5e92a334c4cf88f99ac066073f875.

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
