# Peer review of "Hypoxia as a Stimulus for the Maturation of Meniscal Cells: Highway to Novel Tissue Engineering Strategies?"

_ijms, 2021, doi:10.3390/ijms22136905_

Round 1

Reviewer 1 Report

The authors have investigated the influence of hypoxia (1% O2) on neonatal porcine meniscus for a period of 14 days culture, examining the morphology, tissue structure, gene expression and GAG analysis. The results demonstrate that hypoxia cultured menisci showed that there was greater SOX9 and collagen II staining compared to normoxia cultured menisci, whilst there was a more rounded  cell morphology shown in the inner zone and fibroblastic cell morphology in the outer zone of the menisci. There was found to be increased GAG content after 14 days under hypoxia, whilst COL2A1, SOX9 and HIF-1α gene expression were also upregulated under this condition.

The results demonstrate the beneficial effect, particularly with respect to staining and gene expression data. Additionally, the culture of whole menisci is a novel methodology described in this manuscript.

The authors need to address the following prior to publication: -

  1. For the gene expression data, did you examine ACAN within the samples ? There appear to be no differences in the GAG data between the oxygen tensions, thus on a gene expression level are there differences ?
  2. Do the authors have protein data for collagen II deposition within these menisci ? The immunohistochemical data indicates greater collagen II content under hypoxia , although there appear to be no differences between oxygen conditions at the same time point.
  3. It is difficult to understand the differences between groups with the labelling used for significance. Please change with a line and star to indicate differences between groups.
  4. For the HIF-1α expression, normally this is completed via western blot analysis. Do the authors have any data regarding protein expression of HIF-1a and downstream mechanisms controlling cellular response of meniscus tissue under hypoxia, as detailed in your conclusion ?
  5. The no differences between groups appears to indicate donor variation. It is advised that the authors show the difference porcine donors, as previous data has shown that donor dependent response under hypoxia. Perhaps showing the data with respect to donor may demonstrate greater differences between groups. 

Reviewer 2 Report

This work evaluated the role of hypoxia in the maturation of neonatal meniscal tissue. The reported results are consistent, and they suggest that hypoxia might be exploited in cartilage tissue engineering. The paper is likely to elicit the interest of a broad readership. 

In my opinion, the sole major flaw of this manuscript is related to the quality of Figures 2, 3, and 5; certain portions of these images are blurred, and/or the letters are pixelated. These pictures look like a low-quality JPEG image; I would ask for a higher resolution TIFF image instead. Of course, it might happen that the images were downsampled during the generation of the PDF file of the manuscript and the uploaded images are fine. I leave this aspect to the technical editors to assess.   

Minor comments: 

Line 27: 
The adverb "considerably" seems to be awkward in this sentence. Instead of "opens considerably", I would write "opens new" or "opens exciting". 

Line 81: Instead of "each time points" write "each time point". 

Line 96, Figure 1: The scale bar placed on on pannel A is too thin and contains illegible annotations. Instead, I would provide a more visible scale bar with no text on it.

Line 102: Instead of "the double" write "double". 

Line 217: Instead of "did not separated", please write "did not separate". 

Line 220: 
Instead of "Differently from humans", I would write "Unlike humans". 

Line 228: Instead of "hypoxia preserves", please write "hypoxia preserved". 

Line 297: Instead of "n: 30" I would write "n = 30".

Round 2

Reviewer 1 Report

The authors have addressed my comments appropriately.

Author Response

We thank the reviewer. According to his/her comments, no further changes are needed.